

# Association of the receptor for advanced glycation end-products (RAGE) gene polymorphisms in Malaysian patients with chronic kidney disease

Foo Nian Wong[1], Kek Heng Chua[1], Umah Rani Kuppusamy[1], Chew Ming Wong[2], Soo Kun Lim[2] and Jin Ai Mary Anne Tan[1]

[1] Department of Biomedical Science, Faculty of Medicine, University of Malaya, Kuala Lumpur, Malaysia
[2] Department of Medicine, Faculty of Medicine, University of Malaya, Kuala Lumpur, Malaysia

Corresponding author
Jin Ai Mary Anne Tan,
maryanne@um.edu.my

## ABSTRACT

**Background:** Chronic kidney disease (CKD) is a condition associated with progressive loss of kidney function and kidney damage. The two common causes of CKD are diabetes mellitus and hypertension. Other causes of CKD also include polycystic kidney disease, obstructive uropathy and primary glomerulonephritis. The receptor for advanced glycation end-products (RAGE) is a multi-ligand cell surface receptor of the immunoglobulin superfamily and it has been associated with kidney disease in both non-diabetic and diabetic patients. Presently, data on the association between RAGE polymorphisms and CKD in the Malaysian population is limited, while numerous studies have reported associations of RAGE polymorphisms with diabetic complications in other populations. The present study aims to explore the possibility of using RAGE polymorphisms as candidate markers of CKD in Malaysian population by using association analysis.

**Methods:** A total of 102 non-diabetic CKD patients, 204 diabetic CKD patients and 345 healthy controls were enrolled in the study. DNA isolated from blood samples were subjected to genotyping of RAGE G82S, −374T/A, −429T/C, 1704G/T and 2184A/G polymorphisms using real-time polymerase chain reaction (PCR). The 63-bp deletion, a polymorphism in the RAGE gene promoter, was genotyped using conventional PCR method and visualized using agarose gel electrophoresis. The collective frequencies of genotypes with at least one copy of the minor alleles of the four polymorphisms were compared between the non-diabetic CKD patients, diabetic CKD patients and healthy controls.

**Results:** After adjustment of age, gender and ethnic groups in binary logistic regression analysis, the G82S CT + TT genotypes were associated with non-diabetic CKD patients when compared with diabetic CKD patients ($p = 0.015$, OR = 1.896, 95% CI = 1.132–3.176). After further adjustment of CKD comorbidities, the G82S CT + TT genotypes were still associated with non-diabetic CKD patients when compared with diabetic CKD patients ($p = 0.011$, OR = 2.024, 95% CI = 1.178–3.476). However, it cannot be suggested that G82S polymorphism was associated with CKD in non-diabetic patients in this study. This is because there were no significant differences in the frequencies of G82S CT + TT genotypes between non-diabetic CKD patients and healthy controls. In addition, the RAGE −374T/A,

−429T/C, 1704G/T, 2184A/G and 63-bp deletion polymorphisms were also not associated with non-diabetic CKD patients and diabetic CKD patients in this study. **Conclusion:** The G82S, −374T/A, −429T/C, 1704G/T, 2184A/G and 63-bp deletion polymorphisms examined in this study were not associated with chronic kidney disease in the Malaysian patients.

## INTRODUCTION

Chronic kidney disease (CKD) is a general term for heterogeneous renal disorders which is characterized by progressive kidney damage and estimated glomerular filtration rate (eGFR) of less than 60 ml/min/1.73 $m^2$ for three months or more (*Levey et al., 2003*; *Stevens & Levey, 2009*). CKD is an increasing health problem in Malaysia. In 2013, the prevalence of kidney failure patients requiring hemodialysis and peritoneal dialysis was 970 per million population (pmp) and 95 pmp respectively. In Malaysia, the prevalence of kidney failure has doubled over the last decade (*National Renal Registry, 2014*). The two common causes of kidney failure in Malaysia are diabetes mellitus and hypertension, with other causes being polycystic kidney disease, obstructive uropathy and chronic glomerulonephritis (*National Renal Registry, 2014*).

Receptor for advanced glycation end-products (RAGE) is a multi-ligand cell surface receptor of the immunoglobulin superfamily. The receptors bind to advanced glycation end-products (AGEs), certain members of S100/calgranulin, amphoterin, amyloid $\beta$-sheet fibrils, and advanced oxidation protein products (*Kalea, Schmidt & Hudson, 2009*; *Ramasamy, Yan & Schmidt, 2009*). The interaction between RAGE and its ligand triggers signal transduction which results in various cellular effects such as inflammation, oxidative stress, altered gene expression and apoptosis (*Kalea, Schmidt & Hudson, 2009*; *Xie et al., 2013*). Furthermore, RAGE has also been studied in association with pathogenesis of kidney diseases in animal models (*Myint et al., 2006*; *Guo et al., 2008*; *Reiniger et al., 2010*).

The human RAGE gene is located in the major histocompatibility complex (MHC) class III region on chromosome 6p21.3 (*Kalea, Schmidt & Hudson, 2009*). Numerous polymorphisms have been identified in the promoter region, exons and introns of the RAGE gene (*Hudson, Stickland & Grant, 1998*; *Hudson et al., 2001*; *Kanková et al., 2001*). Many studies have reported on the associations of the common RAGE polymorphisms such as G82S, −374T/A, −429T/C, 1704G/T, 2184A/G and 2250G/A with the development of diabetic nephropathy (*Matsunaga-Irie et al., 2004*; *Prevost et al., 2005*; *Kanková et al., 2005*; *Lindholm et al., 2006*). Moreover, several RAGE polymorphisms have also been investigated in association with lupus nephritis (*Martens et al., 2012*).

Given the associations of several RAGE polymorphisms with kidney diseases in the published literature, the present study aims to explore the possibility of using RAGE polymorphisms as candidate markers of CKD in Malaysian population by using

association analysis. This study investigates the G82S, −374T/A, −429T/C, 1704G/T, 2184A/G and 63-bp deletion polymorphisms based on their effects on RAGE expression and function which potentially affect CKD pathogenesis: (i) G82S polymorphism is a missense mutation in exon 3 of the RAGE gene which potentially regulates RAGE function (*Kalea, Schmidt & Hudson, 2009*), (ii) the polymorphisms in the transcriptional start site of RAGE such as −374T/A, −429T/C and 63-bp deletion polymorphisms regulate the transcription of RAGE (*Hudson et al., 2001*), and (iii) the 1704G/T polymorphism and 2184A/G polymorphism may be responsible for alternative splicing that produces endogenous secretory RAGE which is cytoprotective against RAGE ligands (*Yonekura et al., 2003*; *Schlueter et al., 2003*).

## MATERIALS AND METHODS

### Subject recruitment

CKD patients (40–75-year-old) whose eGFR were less than 60 ml/min/1.73 $m^2$ were recruited from University Malaya Medical Centre (UMMC), Kuala Lumpur between September 2013 and November 2014. To further confirm their CKD status, the eGFR of the patients for the past six months were checked to be constantly less than 60 ml/min/1.73 $m^2$. Estimated GFR was determined using the Modified 4-variable Modification of Diet in Renal Disease (MDRD) study equation (*Levey et al., 2006*). The eGFR was measured on the day of recruitment and the average eGFR of each patient group was calculated. Patients with acute kidney injury which is reversible and kidney transplant recipients whose renal function has reverted to satisfying levels were excluded, as this study aims to investigate kidney diseases which are progressive in nature. Diabetic CKD patients were those with type 2 diabetes (n = 204) and non-diabetic CKD patients comprised of patients with hypertension (54), chronic glomerulonephritis (26), obstructive uropathy (9), analgesic nephropathy (7), polycystic kidney disease (4), urate nephropathy (1) and renal tubular acidosis (1). Healthy controls (35–65-year-old, n = 345) were recruited from blood donors without diabetes, hypertension, heart disease and kidney disease. Peripheral venous blood (3–6 ml) was collected in EDTA tubes for DNA analysis.

This study was approved by the Medical Ethics Committee UMMC (reference number: 982.17) in accordance with the Declaration of Helsinki. Verbal and written informed consent were also obtained from all patients and healthy controls before blood collection.

### DNA extraction and genotyping using commercial polymorphism genotyping assays

Blood specimens were kept at −20 °C until DNA extraction. DNA was extracted using an in-house modified salting out procedure (*Miller, Dykes & Polesky, 1988*). Genotyping for the RAGE G82S, −374T/A, −429T/C, 1704G/T and 2184A/G polymorphisms were performed using the TaqMan® Single Nucleotide Polymorphism (SNP) Genotyping Assays (Life Technologies, Carlsbad, CA, USA) (Table 1) according to the manufacturer's instructions. Briefly, 1 μl of diluted DNA sample, 5 μl of 2X TaqMan® GTXpress™ Master Mix (Life Technologies, Carlsbad, CA, USA), 0.5 μl of 20X TaqMan® SNP
**Table 1 RAGE G82S, −374T/A, −429T/C, 1704G/T and 2184A/G polymorphism identification, assay identification and location.**

| Polymorphism | Polymorphism ID | Assay ID* | Location |
|---|---|---|---|
| G82S | rs2070600 | C__15867521_20 | Chr.6: 32151443 |
| −374T/A | rs1800624 | C___3293837_1_ | Chr.6: 32152387 |
| −429T/C | rs1800625 | C___8848033_1_ | Chr.6: 32152442 |
| 1704G/T | rs184003 | C___2412456_10 | Chr.6: 32150296 |
| 2184A/G | rs3134940 | Custom assay | Chr.6: 32149816 |

**Note:**
* Assay ID of TaqMan® SNP Genotyping Assay (Life Technologies, Carlsbad, CA, USA).

Genotyping Assays and 3.5 μl of DNase-free double-distilled water were mixed to obtain a 10 μl SNP genotyping reaction for each DNA sample.

Genotypes were determined by real-time polymerase chain reaction (PCR) using an Applied Biosystems Fast 7500 Real-Time Thermal Cycler. The pre-PCR stage was 60 °C for 1 min prior to the holding stage at 95 °C for 20 s. PCR conditions include 40 cycles of denaturation at 95 °C for 3 s followed by annealing and elongation at 60 °C for 30 s with a final post-PCR stage at 60 °C for 1 min. Results were analyzed using the Applied Biosystems 7500 Fast Real-Time PCR System and Applied Biosystems TaqMan® Genotyper Software.

## Screening RAGE 63-bp deletion using conventional PCR method

The RAGE 63-bp deletion polymorphism was detected using conventional PCR method. Briefly, 10 μl of final PCR reaction mixture contained 1X DreamTaq Buffer (Thermo Scientific), 0.45 U DreamTaq DNA Polymerase (Thermo Scientific), 0.09 mM deoxyribonucleotide triphosphate (dNTP) (Thermo Scientific), 0.22 μM forward primer, 0.22 μM reverse primer and 80 ng sample DNA. The sequences of forward primer and reverse primer are 5′-GGGGCAGTTCTCTCCTCACT-3′ and 5′-CATGCCTTTGGGA CAAGAGT-3′ respectively.

The PCR was performed with an initial incubation at 94 °C for 5 min, followed by 40 cycles consisting of 94 °C for 30 s, 63.3 °C for 40 s and 72 °C for 40 s. One cycle of final extension at 72 °C for 5 min was programmed to complete the amplification. The PCR products were visualized using electrophoresis on 1.5% agarose gels.

## Statistical analyses

Hardy-Weinberg equilibrium calculator including analysis for ascertainment bias, a web-tool (http://www.oege.org/software/hwe-mr-calc.shtml), was used to assess Hardy-Weinberg equilibrium (HWE) for each RAGE polymorphism using Chi-squared test to examine the differences in genotype distribution between observed and expected frequencies (*Rodriguez, Gaunt & Day, 2009*). The significance of HWE deviation was set at $p < 0.05$.

The statistical power of this unmatched case-control study was estimated using Quanto, version 1.2 (*Gauderman & Morrison, 2001*). In this analysis, the statistical power was calculated for comparing the frequency of genotype with at least one copy of the

minor allele of each RAGE polymorphism between a pair of subject groups. The gene only hypothesis and dominance inheritance mode were selected for power calculation.

Chi-squared test was used to detect the significant difference in each categorical variable. The statistical significance of differences in mean values was analyzed using independent $t$ test or one-way analysis of variance (ANOVA). Binary logistic regression analysis was used to estimate the probability of CKD development attributed to RAGE polymorphism genotypes by adjusting the covariates such as age, gender, ethnic groups and comorbidities of CKD. Odds ratio and 95% confidence interval were calculated. These statistical analyses were performed using the Statistical Package for Social Sciences, version 20 (SPSS Inc., Chicago, IL, USA). The significance level was set at $p < 0.05$.

## RESULTS

The demographic data, eGFR and comorbidities of the study subjects are shown in Table 2. Both non-diabetic CKD ($P_b < 0.001$) (Table 2) and diabetic CKD patients ($P_c < 0.001$) (Table 2) were older compared to the healthy controls while there were no significant differences in categorical variables such as gender and ethnic groups between the study subject groups ($P > 0.05$) (Table 2). Comparison between non-diabetic CKD and diabetic CKD patients showed significant difference in the eGFR levels ($P_a = 0.001$) (Table 2). The CKD patients were also affected with comorbidities such as hypertension, dyslipidemia, ischemic heart disease and stroke. Chi-squared analyses showed that there were significantly more diabetic CKD patients with hypertension ($P_a = 0.043$) (Table 2) and ischemic heart disease ($P_a = 0.002$) (Table 2) than non-diabetic CKD patients.

The collective frequencies of genotypes with at least one copy of the minor alleles of G82S, −374T/A, −429T/C, 1704G/T, 2184A/G and 63-bp deletion polymorphisms in the Malaysian CKD patients and healthy controls are shown in Table 3. Genotype distributions of the six polymorphisms in the non-diabetic CKD, diabetic CKD and healthy controls were in HWE except for the non-diabetic CKD patients with −429T/C polymorphism, diabetic CKD patients with G82S and 1704G/T polymorphisms as well as health controls with 2184A/G polymorphism (Supplementary Information, Table S1).

The frequencies of the genotypes with at least one copy of the G82S T allele (CT + TT genotypes) were significantly higher in the non-diabetic CKD patients than in the diabetic CKD patients (Table 3). Overall, 38.2% of the non-diabetic CKD patients carried at least one copy of the G82S T allele. After adjustment of age, gender and ethnic groups, binary logistic regression analysis indicated that the G82S CT + TT genotypes were associated with non-diabetic CKD patients as compared with diabetic CKD patients ($P = 0.015$, OR = 1.896, 95% CI = 1.132–3.176) (Table 3, Model 1). Further adjustment of CKD comorbidities showed that the G82S CT + TT genotypes were still associated with non-diabetic CKD patients as compared with diabetic CKD patients ($P = 0.011$, OR = 2.024, 95% CI = 1.178–3.476) (Table 3, Model 2). However, the G82S CT + TT genotypes were not associated with non-diabetic CKD patients and diabetic CKD patients as compared with healthy controls ($P > 0.05$) (Table 3).

The genotypes consisting the minor alleles of RAGE −374T/A, −429T/C, 1704G/T, 2184A/G and 63-bp deletion polymorphisms were also not associated with non-diabetic

**Table 2** Demographic data, estimated glomerular filtration rate and comorbidities of non-diabetic chronic kidney disease (CKD) patients, diabetic CKD patients and healthy controls.

| | Non-diabetic CKD patients n = 102 | Diabetic CKD patients n = 204 | Healthy controls n = 345 | $P_a$ | $P_b$ | $P_c$ |
|---|---|---|---|---|---|---|
| Age (years) | 62.98 ± 8.70 | 64.61 ± 7.44 | 43.86 ± 6.30 | 0.141 | < 0.001 | < 0.001 |
| Gender (male/female) | 66/36 | 129/75 | 204/141 | 0.801 | 0.311 | 0.341 |
| Ethnic groups (Malay/Indian/Chinese) | 33/51/18 | 84/74/46 | 127/146/72 | 0.071 | 0.387 | 0.373 |
| eGFR at recruitment (ml/min/1.73 m$^2$) | 32.51 ± 14.51 | 27.04 ± 11.81 | Not available | 0.001 | – | – |
| Hypertension (%) | 73.5 | 83.3 | Not available | 0.043 | – | – |
| Dyslipidemia (%) | 30.4 | 26.0 | Not available | 0.415 | – | – |
| Ischemic heart disease (%) | 11.8 | 27.0 | Not available | 0.002 | – | – |
| Stroke (%) | 3.9 | 5.9 | Not available | 0.468 | – | – |

Note:
Data are expressed as mean ± standard deviation, except for categorical variables, which are reported as numbers or percentages. $P_a$, P-value for non-diabetic CKD patients versus diabetic CKD patients; $P_b$, P-value for non-diabetic CKD patients versus healthy controls; $P_c$, P-value for diabetic CKD patients versus healthy controls.

**Table 3** Association of RAGE G82S, −374T/A, −429T/C, 1704G/T, 2184A/G and 63-bp deletion polymorphisms with chronic kidney disease in Malaysian population.

| Polymorphism | Allele 1/2 | Subjects | Genotype 12 + 22 n (%) | Comparison | Model 1 P | Model 1 OR (95% CI) | Model 2 P | Model 2 OR (95% CI) |
|---|---|---|---|---|---|---|---|---|
| G82S | C/T | ND-CKD | 39 (38.2) | ND-CKD vs D-CKD | 0.015 | 1.896 (1.132–3.176) | 0.011 | 2.024 (1.178–3.476) |
| | | D-CKD | 51 (25.0) | ND-CKD vs HC | 0.749 | 1.135 (0.522–2.467) | 0.598 | 1.440 (0.372–5.576) |
| | | HC | 90 (26.1) | D-CKD vs HC | 0.128 | 0.533 (0.237–1.199) | 0.815 | 1.207 (0.249–5.863) |
| −374T/A | T/A | ND-CKD | 22 (21.6) | ND-CKD vs D-CKD | 0.644 | 0.873 (0.490–1.554) | 0.430 | 0.787 (0.435–1.425) |
| | | D-CKD | 48 (23.5) | ND-CKD vs HC | 0.408 | 1.480 (0.585–3.743) | 0.642 | 1.451 (0.302–6.967) |
| | | HC | 83 (24.1) | D-CKD vs HC | 0.294 | 1.556 (0.682–3.551) | 0.113 | 4.261 (0.710–25.567) |
| −429T/C | T/C | ND-CKD | 16 (15.7) | ND-CKD vs D-CKD | 0.134 | 0.617 (0.328–1.160) | 0.110 | 0.588 (0.306–1.128) |
| | | D-CKD | 46 (22.6) | ND-CKD vs HC | 0.735 | 0.848 (0.326–2.205) | 0.616 | 1.533 (0.288–8.159) |
| | | HC | 81 (23.5) | D-CKD vs HC | 0.103 | 1.948 (0.874–4.342) | 0.054 | 5.727 (0.972–33.731) |
| 1704G/T | G/T | ND-CKD | 42 (41.2) | ND-CKD vs D-CKD | 0.284 | 1.309 (0.799–2.143) | 0.208 | 1.385 (0.835–2.299) |
| | | D-CKD | 70 (34.3) | ND-CKD vs HC | 0.619 | 1.217 (0.562–2.636) | 0.490 | 0.604 (0.144–2.527) |
| | | HC | 125 (36.2) | D-CKD vs HC | 0.799 | 1.100 (0.529–2.287) | 0.834 | 0.853 (0.193–3.763) |
| 2184A/G | A/G | ND-CKD | 16 (15.7) | ND-CKD vs D-CKD | 0.134 | 0.617 (0.328–1.160) | 0.110 | 0.588 (0.306–1.128) |
| | | D-CKD | 46 (22.6) | ND-CKD vs HC | 0.725 | 0.842 (0.324–2.189) | 0.618 | 1.530 (0.287–8.144) |
| | | HC | 82 (23.8) | D-CKD vs HC | 0.105 | 1.941 (0.871–4.324) | 0.054 | 5.725 (0.971–33.741) |
| 63-bp deletion | _/del | ND-CKD | 4 (3.9) | ND-CKD vs D-CKD | 0.735 | 1.244 (0.352–4.392) | 0.863 | 1.120 (0.308–4.069) |
| | | D-CKD | 7 (3.4) | ND-CKD vs HC | 0.273 | 0.303 (0.036–2.560) | 0.387 | 0.268 (0.014–5.283) |
| | | HC | 24 (7.0) | D-CKD vs HC | 0.100 | 0.232 (0.041–1.320) | 0.423 | 0.312 (0.018–5.393) |

Note:
Model 1 adjusted for age, gender and ethnic groups. Model 2 extended Model 1 by also adjusting for comorbidities of CKD such as hypertension, dyslipidemia, ischemic heart disease and stroke. Abbreviations in the table–P, P-value; OR, odd ratio; CI, confidence interval; ND-CKD, non-diabetic CKD; D-CKD, diabetic CKD; HC, healthy control.

CKD patients and diabetic CKD patients ($P > 0.05$) (Table 3). Figure 1 shows the representative gel image of electrophoresed PCR products with or without 63-bp deletion. In this study, the study subjects with 63-bp deletion are heterozygous for the

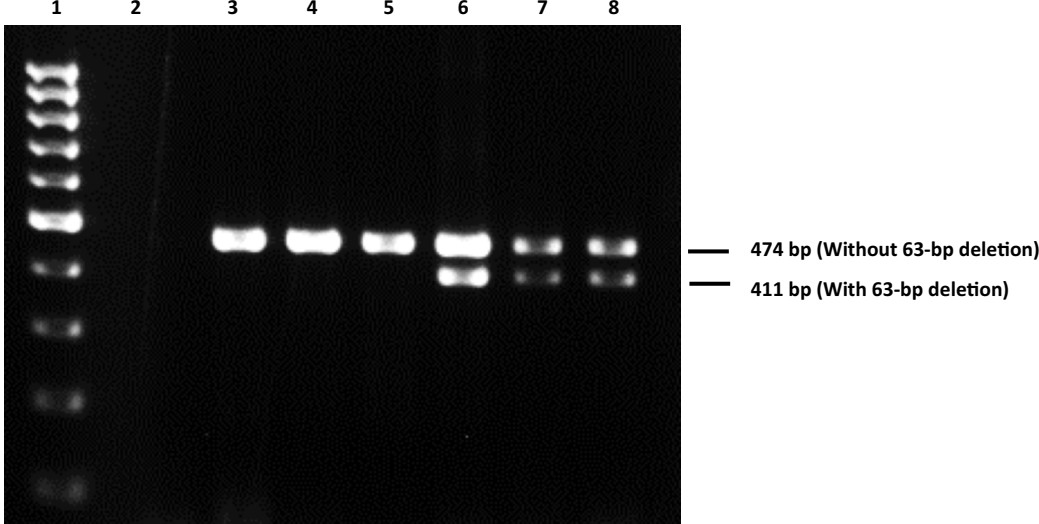

**Figure 1 Representative gel image of electrophoresed PCR products with or without 63-bp deletion.** Lane 1: 100 bp DNA ladder; lane 2: non-template blank; lanes 3–5: selected DNA samples without 63-bp deletion (containing a 474-bp band); lanes 6–8: selected DNA samples which are heterozygous for 63-bp deletion (containing a 474-bp band corresponding to sequence without 63-bp deletion and a 411-bp band corresponding to sequence with 63-bp deletion).

polymorphism and none of the study subjects are homozygous for this particular deletion.

## DISCUSSION

Published studies have corroborated the role of RAGE in the development of diabetes-associated renal diseases; for example, inhibition of RAGE through pharmacological antagonism or gene deletion showed significant improvements in the pathological features of diabetic nephropathy in animal models (*Wendt et al., 2003*; *Reiniger et al., 2010*). In addition to diabetic nephropathy, RAGE is also associated with the pathogenesis of non-diabetic renal diseases, for example, podocyte stress and glomerulosclerosis were elicited in doxorubicin-treated mice, but these features were decreased in the homozygous RAGE-null mice treated with doxorubicin (*Guo et al., 2008*). In view of the potential pathological role of RAGE in kidney diseases, the associations of RAGE gene polymorphisms with CKD were investigated in this study as the polymorphisms may be fundamentally important in CKD development.

In addition to renal diseases, previous human population studies have shown that RAGE polymorphisms were associated with cardiovascular diseases such as ischemic heart disease and stroke (*Zee et al., 2006*; *Poon et al., 2010*). In order to investigate accurately the association of RAGE polymorphisms with CKD, the comorbidities such as ischemic heart disease, stroke and their risk factors (hypertension and dyslipidemia) were adjusted in binary logistic regression analysis to eliminate their confounding effects.

In the present study, the G82S CT + TT genotypes were associated with non-diabetic CKD patients when compared with diabetic CKD patients, but not with healthy controls.

Thus, it cannot be suggested that the T allele is associated with CKD in non-diabetic patients in this study. Studies on the association between G82S polymorphism and kidney disease have showed conflicting results. The frequency of genotype with at least one copy of G82S T allele was significantly higher in type 1 diabetic patients with advanced nephropathy compared with diabetic patients with less severe nephropathy or no nephropathy in France and Belgium (*Prevost et al., 2005*). On the contrary, this polymorphism was not associated with type 1 diabetic nephropathy and lupus nephritis in Denmark and the Netherlands respectively (*Poirier et al., 2001*; *Martens et al., 2012*).

The −374T/A, −429T/C, 1704G/T, 2184A/G and 63-bp deletion polymorphisms did not show any association with non-diabetic CKD and diabetic CKD patients in this study. However, published literature have reported contradictory results for these polymorphisms. The −374T/A polymorphism was associated with diabetic complications including diabetic nephropathy in a Swedish population (*Lindholm et al., 2006*) and has also been implicated in more rapid decline of renal function in Italian CKD patients (*Baragetti et al., 2013*). In contrast, this polymorphism was protective against albumin excretion and cardiovascular disease in type 1 diabetic Finnish patients (*Pettersson-Fernholm et al., 2003*).

Only limited reports are available on the associations of the RAGE −429T/C, 1704G/T, 2184A/G and 63-bp deletion polymorphisms with kidney diseases. In a linkage disequilibrium analysis, the frequency of a haplotype containing the −429T/C and 2184A/G polymorphisms were significantly higher in type 2 diabetic nephropathy (*Kanková et al., 2005*). Furthermore, the 1704G/T polymorphism showed a significant association with type 2 diabetic patients developing nephropathy in a Japanese population (*Matsunaga-Irie et al., 2004*). The 2184A/G polymorphism was associated with increased risk for diabetic nephropathy in type 2 diabetic patients from Central Europe (*Kaňková et al., 2007*) but this polymorphism was reported to play a protective role against diabetic nephropathy in Chinese type 2 diabetic patients (*Cai et al., 2015*). Another study showed that the type 2 diabetic patients with the 63-bp deletion may be protected from diabetic nephropathy, but the type 1 diabetic patients with 63-bp deletion were at risk of diabetic nephropathy in a German population (*Rudofsky et al., 2004*). In addition to diabetic nephropathy, the −374T/A, −429T/C and 2184A/G polymorphisms were also associated with lupus nephritis, worsened proteinuria and decreased renal function in a Dutch population (*Martens et al., 2012*).

Although the RAGE polymorphisms investigated in this study were not associated with CKD, published studies have demonstrated that some of these polymorphisms are associated with altered expression and function of RAGE which may underlie disease development. Several lines of evidence have suggested the importance of the G82S T allele in the mediation of RAGE-ligand binding and activation of downstream signaling pathways. The G82S T allele renders higher affinity of RAGE towards ligands such as AGEs and S100/calgranulin (*Hofmann et al., 2002*; *Osawa et al., 2007*). Another study showed that the RAGE protein with G82S polymorphism promoted $N$-linked glycosylation at the Asparagine 81 site, which was associated with increased ligand binding and pro-inflammatory NF-$\kappa$B activation (*Park, Kleffmann & Hessian, 2011*). The deleterious cellular effects resulting from increased ligand binding such as oxidative stress and

inflammation (*Kalea, Schmidt & Hudson, 2009*) are fundamental to CKD development (*Ruiz et al., 2013*).

The RAGE −374T/A, −429T/C and 63-bp deletion polymorphisms which occur in the transcriptional start site of RAGE gene may affect the transcriptional regulation. A published study showed that these polymorphisms resulted in an increase of transcriptional activity (*Hudson et al., 2001*), indicating an enhanced expression of RAGE which is associated with the progression and severity of renal dysfunction (*Wendt et al., 2003*; *Hou et al., 2004*). However, the functional impact of 1704G/T and 2184A/G polymorphisms remain to be elucidated.

Despite the evidence of RAGE polymorphisms associating with kidney diseases, the selected RAGE polymorphisms were not associated with CKD in the present study. A plausible explanation for this discrepancy is the ethnic and regional differences between Malaysian and other populations. Malaysia is a unique country because of its multiracial population which includes the Malays, Chinese and Indians who comprise the majority of the population. These ethnic groups may not be genetically similar to other populations worldwide. Two published reports have provided an example of this concept that the association of 2184A/G polymorphism with type 2 diabetic nephropathy in Central Europe (*Kaňková et al., 2007*) is contradictory to the decreased risk of type 2 diabetic nephropathy in the Chinese (*Cai et al., 2015*). This highlights the possible influence of ethno-regional difference on the association between polymorphism and disease. In addition, RAGE polymorphisms may not be CKD-specific in Malaysian patients according to the findings in this study. Given the conflicting findings in the association studies, the relationship between RAGE polymorphisms and kidney diseases should be interpreted with caution.

The main limitation of this study was the subjects who were recruited only from a medical center may not be a representative of the general population. Current sample size determined in a priori power analysis could detect significant associations with 80% statistical power. During final analysis, however, most of the comparisons were associated with low statistical power (Table S2). The low statistical power could be attributed to the odds ratio which is very close to the null value, small sample size and low frequency of risk allele because statistical power is influenced by effect size, sample size and disease allele frequency (*Gordon, 2005*; *Schneider, 2013*). The low statistical power in this study is likely due to the low risk allele frequencies. For example, the risk allele frequencies of several RAGE gene polymorphisms such as −374T/A, −429T/C and 2184A/G are lower in the Malaysian populations compared with the Caucasian populations (*Kanková et al., 2005*; *Kalousová et al., 2007*).

*Baragetti et al. (2013)* demonstrated in a prospective study that the A allele of −374T/A polymorphism was associated with more rapid renal function decline. This suggests that RAGE polymorphisms may affect CKD progression over a longer time span. Therefore, the relationships between RAGE polymorphisms and CKD can be investigated using prospective study design to validate the negative findings in the current study. Furthermore, the polymorphisms of other genes such as transforming growth factor-$\beta$1

(TGF-$\beta$1), non-muscle myosin heavy chain 9 (MYH9) and apolipoprotein L1 (APOL1) have been shown in associations with kidney diseases (*Freedman et al., 2009*; *Vuong et al., 2009*; *Langefeld et al., 2015*). These candidate genes would serve as promising tools for CKD marker discovery.

## CONCLUSION

Based on the current findings, the RAGE G82S, −374T/A, −429T/C, 1704G/T, 2184A/G and 63-bp deletion polymorphisms are not qualified to be the markers of CKD in Malaysian populations on the account of the null associations between these polymorphisms and CKD. Therefore, it may not be useful to predict CKD using RAGE polymorphisms in Malaysian patients.

### Funding

This research is supported by Fundamental Research Grant Scheme (FRGS) (Project no. FP032-2014A) from the Ministry of Higher Education, Malaysia and Postgraduate Research Grant (Project no. PG031-2014B) from University of Malaya. The funders had no role in study design, data collection and analysis, decision to publish, or preparation of the manuscript.

### Competing Interests

The authors declare there are no competing interests.

### Author Contributions

- Foo Nian Wong conceived and designed the experiments, performed the experiments, analyzed the data, wrote the paper, prepared figures and/or tables.
- Kek Heng Chua conceived and designed the experiments, contributed reagents/materials/analysis tools.
- Umah Rani Kuppusamy analyzed the data, contributed reagents/materials/analysis tools.
- Chew Ming Wong reviewed drafts of the paper.
- Soo Kun Lim reviewed drafts of the paper.
- Jin Ai Mary Anne Tan conceived and designed the experiments, contributed reagents/materials/analysis tools.

### Human Ethics

The following information was supplied relating to ethical approvals (i.e., approving body and any reference numbers):

This study was approved by the Medical Ethics Committee UMMC (reference number: 982.17) in accordance with the Declaration of Helsinki. Verbal and written informed consent were also obtained from all patients and healthy controls before blood collection.

### Data Deposition

The raw data has been supplied as Supplemental Dataset Files.

## Supplemental Information

Supplemental information for this article can be found online at http://dx.doi.org/10.7717/peerj.1908#supplemental-information.

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
