# Peer review of "Association of the receptor for advanced glycation end-products (RAGE) gene polymorphisms in Malaysian patients with chronic kidney disease"

_PeerJ, doi:10.7717/peerj.1908_

## Round 0.1 · original submission · Major Revisions

· Academic Editor

Major Revisions

Please, carefully address the points raised by the reviewers. In particular, correct existing errors with the references and include missing relevant works as suggested by both reviewers. Also, the discussion of the results should be improved by providing reasoned dissertation about the findings, their novelty and validity. Probably by comparing the frequencies of the G82S polymorphism in other populations.

·

Basic reporting

There are some minor comments:

1. The authors need to check the language, grammars and punctuation marks (commas, etc). For example:
a. Introduction sentence #1 --> When the authors rephrase a definition from other paper, the new definition should be precisely correct: ”irrespective of clinical diagnosis”, ”and/or”
b. Introduction sentence #2 seems to have no importance to mention. I suggest to remove it.
c. Introduction sentence #5: ”Malaysia has seen…” --> human can see, a country cannot see.
d. ”The two common causes …, with other causes including…” --> lack of references
e. B-sheet fibrils --> use proper "Beta" symbol.

2. Introduction, paragraph 2, line 81: It is, perhaps, a quite strong statement to say that RAGE plays "an important role" in severity of kidney diseases, while it is merely associated according to the D’Agati’s and Martens’s articles.

3. Introduction, paragraph 3, line 101-102: two exactly same references were written in one sentence.

4. Introduction, last paragraph: The authors need to rephrase this paragraph to clearly mention that there are several articles investigating association between RAGE SNPs and CKD in non-Malaysians (and perhaps in Singaporeans), provide some references, and then the authors can proceed with the statement that no studies have been carried out in Malaysian population.

5. Materials and Methods, line 152: The authors need to cite Quanto by citing the QUANTO documentation (technical report)

Experimental design

Another minor comment:
Materials & Methods, line 119-121: It is a common knowledge for medical practitioners but the author might still need to mention why acute CKD patients and patients with kidney transplant were not included in the experiment, as some readers might have statistics background without prior knowledge on the diseases.

Validity of the findings

Minor comments:

1. The conclusion should also mention:
a. The association between overall Malaysian CKD and G82S SNP.
b. That the association does not apply to the Malaysian Chinese populations.
c. That the association does not apply to the other 3 SNPs.

2. A paragraph should start with the important idea of why the paragraph needs to be written, and then followed by additional information.
a. Discussion, paragraph 2, line 213: The first sentence of the paragraph should mention the previous association studies between G28S and CKD. Association study between G82S and diabetic retinopathy should be explained later in the paragraph. Carefully rearranging the sentences in this paragraph might be useful.
b. Discussion, paragraph 3, line 227: The paragraph should start with the statement that D-CKD is quite related with soluble RAGE, and ND-CKD too. Then start explaining why.

3. The authors might need to consider mentioning that CKD for Chinese population is associated with other SNPs, such as ABCB1 C3435T.

Major comments:

1. Discussion, paragraph 2, line 225: "The conflicting findings regarding G82S SNP association is related to differences between Caucasian and Southeast Asian populations".

When providing a conjecture or a reason, the authors should provide additional proof(s) and reference(s).

2. Lack of explanation in the Discussion:
a. The authors need to provide reason(s), why Malaysian Indians and Malays have an association between G28S SNP and CKD, and bring the proof that might support this reason. The author needs to provide reason(s) for the Chinese population, too.

b. The authors need to provide a reason, why the frequency of the minor T allele of G82S SNP in ND-CKD was not quite different to D-CKD in the Indian populations. Do not forget to provide proper conjecture and supporting proofs.

Additional comments

In this article the authors proposed an association between four RAGE SNPs and diabetic+non diabetic CKD on three Malaysian races. The manuscript is considered novel and can potentially contribute to the public database of SNPs such as SNPedia.

Reviewer 2 ·

Basic reporting

Looking at reference Matsunaga-Irie et al., 2004, it refers to the 242T allelle, which is not one of the RAGE SNPs referenced to in the text.

In line 95, it is stated that an association between RAGE SNPs and kidney disease has been reported, but references are missing. Only specific diabetic nephropathy references are listed.

Experimental design

Methods section:
• In methods, it needs to be specified how many times eGFR was measured during the three months and how the significance was calculated. All measurements during the three months should be in supplementary. Is it the mean eGFR given in the supplementary? This should be clearer.

• It is concerning that the age is ~20 years younger in the healthy control cohort and that the Fischer’s exact test does not take this into consideration. Instead one could use logistic regression to include covariates such as age.

• It is not stated what the HWE webtool is called. It would also be good to provide a link. Moreover, a reference to webtool Quanto v1.2 is missing.

• A more in-depth description about how HWE web-tool and Quanto were applied and why would be good.

• Supplementary material of the output from these webtools is missing.

Validity of the findings

• One of my concerns is that the chosen SNPs are not CKD-specific. Hence, the G82S polymorphism might as well be a risk allele associated with one of many comorbid diseases that cause CKD. You should look into what other comorbid diseases the CKD patients have.

• Due to the small sample sizes in the ethnic-specific analysis, an estimate of the statistical power should be made, which gives the probability of false positives.

Additional comments

In general, it is a nicely written and simple paper, which has a clear goal and statement.

---

## Round 0.2 · Major Revisions

· Academic Editor

Major Revisions

Please, address the comments of the reviewers.

I appreciate the substantial improvement in the flow of the information, particularly in the presentation of results and discussion respect to the previous version of the manuscript. Nevertheless, these two sections might benefit from some editing. Please, describe the results in more detail and try to use clearer statements. Given that this work does not identify positive associations and that these results might be controversial (and therefore of interest) respect to published literature, I would appreciate that a more detailed, critical discussion is included. In particular, a thorough discussion on the statistical power to detect the polymorphisms and any experimental artefacts that might hinder detection and association, as suggested by reviewer 2.

Furthermore, you might want to consider reporting the actual p-values rather than the significance level at which the difference is evaluated (For example, p-value<0.05 line 163, among many others).

·

Basic reporting

The language, grammar, punctuation marks, and flows are better in this revised version, especially on the Introduction. However, the story flows in the result, discussion, and conclusion are still confusing. For example:

Which table must the readers refer to when the "p value" or "significant different" is mentioned in these sentences below?
1. Result, paragraph 1, sentence 2: "Both ND-CKD and D-CKD were older (p<0.05)!
2. Result, paragraph 1, sentence 2: "While there were no significant differences in gender and ethnic groups"
3. Result, paragraph 1, sentence 3: "Comparison between ND-CKD and D-CKD patients showed significant difference in the eFGR levels"
4. Result, paragraph 1, last sentence: "Chi-squared analysis showed that..."

Experimental design

No Comments

Validity of the findings

The article concludes that these four proposed RAGE SNPs are not associated with CKD. The further studies mentioned in the conclusion (i.e. larger samples or add more SNPs) MUST be conducted in this article. I suggest go wider by adding more SNPs or observing the whole genome, because adding more patients could be costly.

Additional comments

In this article the authors proposed an association between four RAGE SNPs and CKD on Malaysians. The analyses and rationale have been delivered quite well. However, the authors found no strong association. My suggestion is that the authors increase the sample size or add more candidate SNPs, until they find something outstanding/useful for the public SNP dictionary, such as SNPedia, and then resubmit the revised article. Otherwise, the importance of this article becomes very low.

Reviewer 2 ·

Basic reporting

Line 163: You should shortly explain why you performed the comorbidity analysis.

Line 249: "to be not associated with..." should be changed to "not to be associated with..."

In the discussion section you should include more discussion on why you did not find any associations in this study e.g. study design, sample size, etc. The discussion focuses too much on what others have done. Instead you should point out limitations from the study and propose how it could be done in future studies if one had other ressources.

Experimental design

Fine

Validity of the findings

Fine

---

## Round 0.3 · Major Revisions

· Academic Editor

Major Revisions

Please, address the comments by the referee. The article has improved substantially although there might be need for some additional work in the Discussion. In particular, in the contextualisation of the results, as the referee points out.

·

Basic reporting

The authors' current flow in introducing the objective in Introduction section is: "Many studies have associated RAGE SNPs to CKD. No study has been done with Malaysian patients study case. So, we're doing that." The objective of the article should be modified to "evaluating potential benefits of the promising RAGE SNPs analyses in Malaysian CKD patients" (which, later based on results, "negates" the previous association studies, perhaps in Malaysians case). Thus, the logic in Introduction should be revised. Please revise the Introduction flow. Hint: the 1st and 2nd paragraphs in the Discussion are very good for inspiration. The authors have written a very good logical flow on the first paragraph of Discussion.

Experimental design

No Comments

Validity of the findings

Discussion:
The authors have addressed their limitation, which is very well written. Authors must also bring some extra reasonings in relation to the contradictions in the Discussion. Why did the previous studies mention that there is an association between RAGE SNPs and CKD? Why do the authors find the absence of association? Because of ethnic-specific genotype? Or due to low statistical power? Or other reasons too? There could be wild but educated scientific conjectures. Then, elaborate these conjectures after reading some other references.

Conclusion:
A suggestion to the logic in the conclusion:
- Is the RAGE SNPs presence/absence in Malaysian CKD patients?
- Can we say that there are no association between RAGE SNPs and CKD?
- Predicting CKD (or targeting a CKD therapy) using RAGE SNPs might or might not be useful?

Please emphasize on "not recommending" readers to use RAGE SNPs to predict CKD. Do not stress too much on recommending readers to choose other SNPs.

Additional comments

The authors should now emphasize on contradicting their finding with previous studies on associating RAGE SNP with CKD. This is another way of keeping the article to have high impact on the research area. Given six polymorphism candidates now, there is no need to add more wet lab experiments. Authors is almost there and may now focus on finding good references for more reasonings in discussion.

---

## Round 0.4 · accepted · Accept

· Academic Editor

Accept

Having addressed the comments raised by the reviewer (which you can do in production at this point), I have no further objections to the publication of the article.

·

Basic reporting

1. Please explain in 1-2 sentences to the reader about Malaysian ethnicity (i.e. that Malaysian population comprises three ethnics), and that Malaysian population is genotypically similar/quite different to the population of Europe and other regions mentioned in the paper. Especially because ethnic and regional differences were claimed by the authors to be the reason of lack of association.
2. Please check again the text carefully for errors in language, word choice, and grammar.

Experimental design

No Comments

Validity of the findings

No Comments